# Fire Properties of Bed Mattresses Focusing on the Fire Growth Rate and Flame Height

**DOI:** 10.3390/ma15113757

**Published:** 2022-05-24

**Authors:** Jong-Jin Jeong, Masayuki Mizuno, Kye-Won Park, Hyeon-Jin Lim, Chang-Geun Cho

**Affiliations:** 1Department of Architecture Engineering, Chosun University, Gwangju 61452, Korea; jjj018018@gmail.com (J.-J.J.); jericho0220@naver.com (H.-J.L.); chocg@chosun.ac.kr (C.-G.C.); 2Department of Global Fire Science and Technology, Graduate School of Science and Technology, Tokyo University of Science, Chiba 278-8510, Japan; 3Fire Insurers Laboratories of Korea, 1030, Gyeongchung-daero, Ganam-eup, Yeoju-si 469-881, Korea; 25hyun@daum.net

**Keywords:** full-scale combustion, ISO 12949, installation height, fire growth, flame height

## Abstract

Bed mattresses are rated as products to cause a fire hazard because of their very high heat release rate among indoor combustibles. In this study, fire growth rate and flame height were measured through a series of combustion experiments on a full scale in order to provide information regarding mattress fire characteristics. The experiments were conducted in an open space, and bed mattresses as the test samples were installed at different installation heights (0~515 mm). The experiment results revealed that the higher the bed mattress was installed, the higher the fire growth rate, the heat release rate, and the flame height. Additionally, the time of the mattress to reach 1 MW was evaluated as the category “medium” in the NFPA 72 standards. The flame heights showed a good coincidence compared to the existing flame height model equations, proving the applicability of the model to the mattress combustion.

## 1. Introduction

Several studies on the combustion of combustibles have been conducted in various fields, such as architecture, machinery, physics, and chemistry, to understand complex fire phenomena and identify the mechanism of fire behavior. In addition, to prevent fires, it is essential to collect data for quantitative evaluations through many experiments and understand fire properties. For this purpose, basic research has been continuously conducted in developed countries, and many databases have been accumulated. However, over time, the types of combustibles have been diversified, and flame-retardant products for fire prevention have also been developed; therefore, research on combustion and experiments with combustibles are still ongoing.

The fire danger associated with mattresses is determined by the ignitability of components, flame spread and fire growth, and heat release rate during combustion. Moreover, these are affected by the conditions of the smoke layer formed under the ceiling when combusting. In addition, when the temperature of the smoke layer increases, strong incident radiation heat of the combustibles generates flammable gas in the room and causes flashover [1]. According to the White Paper on Fire Service in Japan, bedding, including bed mattresses, accounted for the highest proportion (12.0%) of fire sources that caused deaths in residential fires (108 of 899 casualties), with the exception of 449 casualties (49.9%) with unclear causes. Therefore, it is very important that the combustion behavior of bed mattresses is elucidated through the combustion experiments on a full-scale [2].

In this study, the fire growth rate and flame height were evaluated through a full-scale fire test of the mattress. The fire growth rate is used as important data to predict fire properties, such as the heat release rate and flame-spread speed. In addition, the U.S. National Fire Protection Association (NFPA) classifies mattresses as highly flammable. Flame height is a major physical quantity that is directly related to fire safety in terms of fire spread and fire resistance of building structures. Various methods are used to evaluate the flame height; however, it is not easy to measure it clearly because of the uncertainty of the flame boundary and its geometric deformation. In addition, no standardized method exists, so the flame height is calculated according to various equations [3,4,5]. In this study, Heskestad’s and Zukoski’s models, which are representatively used among standard flame height equations, were used [6,7].

The melting and dripping of the significant quantities of mattress components, such as polyester fiber batting and polyurethane foams, may cause ‘pool’ fires, thereby increasing the threat of fire spread to other items in the room [8]. Here, if the installation height of the mattress is different, it may cause different effects of the pool fire on the burning mattress. Therefore, we investigated the effect of the installation height on the mattress combustion through a series of tests. The aim of this study is to provide information on the fire properties of mattresses to be used as basic data for safety evaluation based on an engineered approach. Specifically, the heat release rate and fire growth rates were evaluated through a series of full-scale combustion experiments with different mattress-installation heights, and the measured flame heights were compared with existing model formulas.

## 2. Full-Scale Fire Test

### 2.1. Experimental Overview

The mattresses used in the combustion experiments were Japanese-manufactured pocket-coil-type mattresses (970 mm wide, 1960 mm long, and 230 mm high) (Figure 1). The mattress was composed of cotton, polyester textiles, urethane foams, and coils; the outer surface was cotton, and the inner surface was polyester, urethane foam, and coils (Figure 2). The cross-sectional composition of the mattress materials was the same on the top and bottom surfaces. In addition, the total weight of the mattress was 25.81 kg; the weight of the fabric material, except for springs and frame, was measured at 6.16 kg.

### 2.2. Experimental Location

The experiment was conducted in an open space using a furniture calorimeter (5 m width × 5 m depth × 2.7 m height). Additionally, fireproof curtains were installed at the bottom of the hood around to prevent leakage of generated smoke and accurately measure the heat release rate. The openings existed below, around 1.5 m in height from the floor (Figure 3).

### 2.3. Mattress Installation Height

In a series of experiments, the installation height of the mattress (HB (mm)) was set at 0 mm, 115 mm, 215 mm, 315 mm, and 515 mm, as shown in Figure 4. In the case of HB 0, calcium-silicate boards were put on the floor, and a mattress was placed on them. In the case of HB 115 mm, a steel frame (whose height was 115 mm) was put on the calcium-silicate board floor, and a mattress was placed on the frame. In the case of HB 215 mm, the concrete blocks (100 mm thick) were installed under four corners of the frame. For HB 315 and 515 mm, the two-tier blocks and four-tier blocks were installed under the frame to adjust the installation height of the mattress.

### 2.4. Ignition Method

The liner burner system for ignition of a mattress (manufacturer: Fire Insurers Laboratories of Korea) was produced according to ISO 12949 [9] and used in the experiment (Figure 5). The fuel of the burner was propane gas (over 99% in component concentration), and the burner system was divided into the top and side burners. The fuel gas was supplied to the top burner at 12.9 L/min (correspond to 18 kW) for 70 s after ignition and the side burner at 6.6 L/min (correspond to 9 kW) for 50 s. The heat release output (27 kW) of the top burner and side burner was similar to the heat release generated by the ignition of blankets, pillows, small dust box, etc.

### 2.5. Measurement Methods

#### 2.5.1. Heat Release Rate and Total Heat Release

The oxygen concentration, flow rate, and temperature of exhaust gas in the duct were measured using a furniture calorimeter, and the heat release rate was calculated by Equation (1), based on the oxygen-consumption method [10]. In addition, the total heat release (THR) was calculated by accumulating the heat release rate:(1)$$\dot{Q}=\frac{E^{1}\dot{m\;}\frac{M_{O_{2}}}{M_{air}}\;\left(1-X_{H_{2}O}^{0}\right)}{\frac{\alpha -1}{X_{O_{2}}^{0}}+\frac{1-\frac{X_{O_{2}}}{1-X_{CO_{2}}}}{X_{O_{2}}^{0}-\frac{X_{O_{2}}\left(1-X_{CO_{2}}^{0}\right)}{1-X_{CO_{2}}}}}$$
where $\dot{Q}$ is heat release rate (kW), *E*^1^ is the amount of energy developed per consumed kilogram of oxygen (kJ/kg), $\dot{m}$ is the mass flow in exhaust duct (kg/s), $M_{O_{2}}$ is the molecular weight of oxygen (g/mol), $M_{air}$ is the molecular weight of air (actually the molar weight of the gas flow in the duct) (g/mol), α is the ratio between the number of moles of combustion products including nitrogen and the number of moles of reactants including nitrogen (expansion factor), $X_{O_{2}}^{0}$ is mole fraction of O_2_ in ambient air, measured on dry gases (−), $X_{CO_{2}}^{0}$ is mole fraction of CO_2_ in ambient air, measured on dry gases (−), $X_{H_{2}O}^{0}$ is the mole fraction of H_2_O in ambient air, $X_{O_{2}}$ is the mole fraction of O_2_ in flue gases, measured on dry gases (−), and $X_{CO_{2}}$ is the mole fraction of CO_2_ in flue gases, measured on dry gases (−).

#### 2.5.2. Flame Height

In the full-scale experiment, video cameras were installed on the cross section of the mattress to measure the flame height. The standard height was recorded by the cameras maintained in each position using a 5 m long bar with 10 cm scale resolution before the experiment. In addition, the flame height was extracted at 30 frames per second, and the average flame was obtained by measuring the continuous and intermittent flames based on each pitch where the flame expanded and contracted with air inflow change by turbulence.

#### 2.5.3. Statistical Analyses

In order to support the experimental results, we performed statistical analysis (single-factor ANOVA) on the heat release rate and the flame height. The results showed statistically significant differences in the installation height when *p* ≤ 0.05.

## 3. Results

### 3.1. Heat Release Rate and Total Heat Release

Figure 6 shows the heat release rates (HRR) and total heat release (THR) values over time for the mattress combustion, with the time of burner ignition at 0 s. The maximum heat release rate and the time required before reaching 1 MW (which is a representative value of fire growth) were compared for different mattress installation heights (HB, in mm). With increasing installation heights, the heat release rates were higher and the fire growth more rapid (515 mm ≈ 315 mm; 315 mm > 215 mm; 215 mm > 115 mm; and 115 mm > 0 mm, in terms of max. HRR and fire growth). The results showed statistically significant differences over time in the installation height, excluding 515 mm (*p* ≤ 0.05). Furthermore, the values of accumulated heat release rate increased with the installation height in the middle of combustion, but the values of the total heat release were 250 MJ at 600 s after ignition for all heights.

### 3.2. Flame Height

Figure 7 shows the flame height for HB 515 mm. After ignition, the mattress surface began to burn, and the melted materials fell to the floor. After 220 s, the flame on the mattress and the flame on the floor had fully integrated. In addition, the flame height gradually increased and reached the maximum.

Figure 8 shows the results of the average flame height (height from the floor) in 20-s increments from ignition based on the video image (they were not measured in the cases of HB 115 mm and 315 mm). There was no significant difference in the average flame height from ignition to 70 s. It was observed that the flame height was almost unchanged until the burner heat reached 70 s, and the fire growth started at 100 s. However, after 100 s, each HB showed different results. For HB 0 mm, the flame height gradually increased after 100 s and reached its maximum (2.07 m) at 260 s. For HB 215 mm, after 100 s, the flame height rose rapidly by about 0.6 m at the point of 120 s and gradually increased to the maximum (2.17 m) at 240 s. For HB 515 mm, it increased exponentially after 100 s and reached the maximum (2.24 m) at 180 s. The higher the mattress installation height, the higher the flame height (*p* ≤ 0.05).

## 4. Analysis of Experimental Results

### 4.1. Analysis of Fire Growth Rate

In the NFPA 72 National Fire Alarm Code, fire growth is classified into four categories: slow, medium, fast, and ultrafast. The time (*t_g_*) for the HRR of the fire to reach 1 MW at each classification is as follows: slow = 300 s to 600 s, medium = 150 s to 300 s, fast = 75 s to 150 s, and ultrafast = 75 s or less. Table 1 lists the classifications of the fire growth time (*t_g_* (s)) and fire growth rate (*α*) defined in NFPA 72 [11,12].

In this experiment, the fire growth of the bed mattresses belongs to the category of “medium” according to the time *t_g_* in the NFPA 72 (Table 1) because the times to reach 1 MW were 275 s, 221 s, 204 s, 175 s, and 178 s for the mattress installation heights HB 0, 115, 215, 315, and 515 mm, respectively (Figure 9). Furthermore, the fire growth rate was obtained as the coefficient of the *t* squared term of the HRR equation *Q =* α*t*^2^ through the point of the maximum HRR and the corresponding time in Figure 9. As a result, the higher the mattress installation height, the larger the fire growth rate. However, as the HRRs of HB 315 and 515 show similar results, if the installation height of the mattress exceeds a certain level, it is thought that the maximum HRR and fire growth rate will no longer increase.

According to the existing literature [13], the mattress is categorized as the “medium” in the fire growth curve. In this experiment, the values of fire growth time and fire growth rate belonged to “medium”, the same as the past classification.

### 4.2. Flame Height Analysis

The combustion of a mattress generated a flame on the mattress for a certain period of time after ignition; as time elapsed, the mattress materials that were melted and fallen apart with the coils began to burn and create a flame on the floor (Figure 7). The flame generated on the floor grew with a progressive increase in the amount of the melted and fallen materials to the floor and merged with the flame on the mattress.

In this study, the Heskestad [6] and Zukoski [7] equations, which are widely used, were applied as the prediction equations for the average flame height, and the experimental data were compared with them. Specifically, Heskestad proposed Equation (2) using the data of experiments conducted in open spaces and compared them with the experimental data by other researchers. In addition, Zukoski proposed Equation (3) using the correlation between the dimensionless heat release rate and the ratio of the average flame height divided by the diameter of the fire source.

In the experiment, the reference point for measuring the flame height was the floor surface, not the top or bottom surface of the mattress, because of the merged flame generated from the burning material falling to the floor during the mattress combustion. In addition, the fire source diameter (D) was calculated by reading the range of the combustion part from the video every 20 s starting from 100 s after ignition and was calculated using these values. In order to measure the length corresponding to the long part of the combustion range, the horizontal axis length of the mattress surface was measured.

Figure 10 shows two comparisons between plots from the experimental data in this study and the lines by Equations (2) and (3). For the Heskestad model, the experimental values are, on average, larger than the line of the model. The plots from HB 0 mm, which means the mattress was positioned on the floor, comparatively match the line from the Heskestad model, but the most plots from HB 215 and 515 mm, where the mattress was installed on the frame above the floor, tend to be larger than the line. However, for the Zukoski model, the experimental values almost correspond to the calculated values from the model. Furthermore, according to the mean squared errors (MSE) of both models in relation to the experimental values, the results of flame height in the full-scale experiment of mattress combustion were more consistent with the Zukoski model than the Heskestad model; however, the applicability of both equations to the experimental values was largely confirmed.
(2)$$\frac{z_{f}}{D}=0.23\frac{Q^{2/5}}{D}-1.02$$
(3)$$\frac{z_{f}}{D}=\{\begin{array}{c} 3.3Q_{D}^{*2/3}\;(Q_{D}^{*}<1.0)\\ 3.3Q_{D}^{*2/5}\;\left(Q_{D}^{*}\geq 1.0\right) \end{array}$$
(4)$$Q_{D}^{*}=\frac{Q}{\rho_{\infty }C_{p\infty }T_{\infty }\sqrt{gD}D^{2}}$$
where z_f_ is the average flame height (m), $Q_{D}^{*}$ is dimensionless heat release rate (−), Q is the heat release rate (kW), D is the fire source diameter (m), $\rho_{\infty }$ is the density of ambient air (kg/m^3^), $C_{p\infty }$ is the specific heat of ambient air (kJ/kg·K), $T_{\infty }$ is the temperature of ambient air (K), and g is the acceleration of gravity (9.81 m/s).

## 5. Conclusions

In this study, the heat release rates and fire growth rates were evaluated through a series of full-scale experiments in mattress combustion with different installation heights and flame heights. The applicability of existing model formulas of flame height to the mattress combustion was confirmed using the experimental data. The results are as follows:(1)In the experiment with the mattress installed 0–515 mm on and above the floor, a trend was observed where the fire growth was faster, and the maximum HRR was larger at greater installation heights. Furthermore, the accumulated HRR over time earlier increased with a rise in installation height. The total heat release values for all the conditions were approximately 250 MJ at 600 s after ignition;(2)The higher the installation height of the mattress, the greater the fire growth rate and the shorter the fire growth time. In addition, as a result of the experiment, fire growth was classified as “medium”, according to the NFPA standard;(3)In descending order, according to the maximum flame height, the installed heights of the bed mattress HB were 515 mm, 215 mm, and 0 mm. The higher the installation height of the mattress was, the higher the flame height. In addition, the applicability of existing flame height model equations to the experimental results of the full-scale bed mattress combustion was largely confirmed. In addition, in measuring the flame height of the mattress, we checked whether it is reasonable to use the floor level (the point where the pool fire is) or the mattress top surface level (the flame origin by the mattress combustion) as the bottom level of the flame height, using the predictive formula. It was confirmed that the experimental results approximately coincided with these prediction results by adopting the floor level as the bottom of the flame.

As stated in Section 3.2, the results of the experiments showed the phenomenon where the components of the mattress melted down and caused the pool fire. This phenomenon was also observed in the existing research [1]. Moreover, the experimental results showed that the higher the installation level of the mattress, the faster the fire rate grows. (However, even if the installation height is 315 mm or more, the results are thought to be similar.) This is thought to be due to the difference in the amount of air inflow under the mattress. Therefore, in order to reduce the danger of the fire when increasing the height of mattress installation, it is necessary to manufacture a frame that blocks air inflow to suppress the combustion of the pool fire.

## Figures and Tables

**Figure 1 materials-15-03757-f001:**
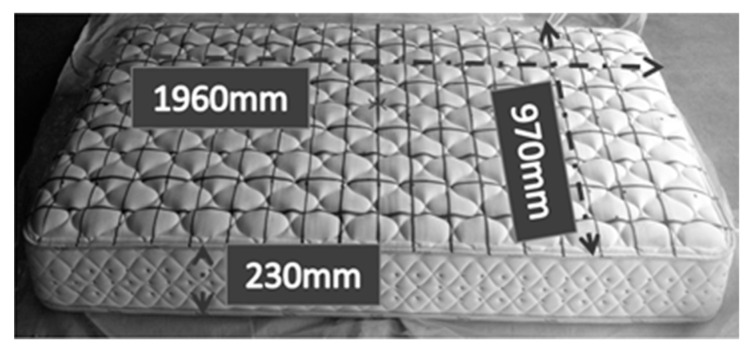
Bed-mattress dimensions.

**Figure 2 materials-15-03757-f002:**
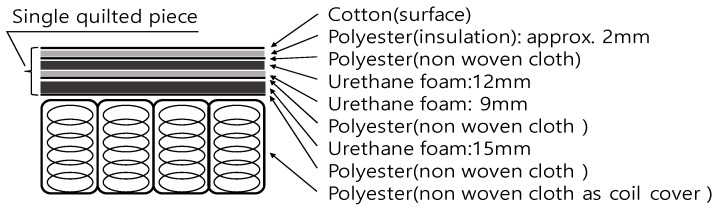
Mattress composition (single side).

**Figure 3 materials-15-03757-f003:**
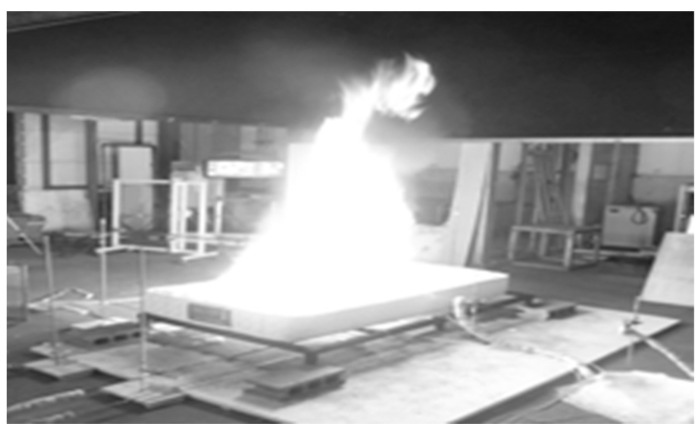
Mattress combustion location.

**Figure 4 materials-15-03757-f004:**
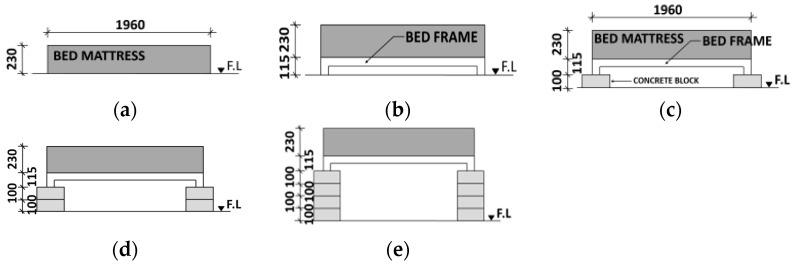
Bed-mattress placement on the floor. (**a**) HB 0 mm. (**b**) HB 115 mm. (**c**) HB 215 mm. (**d**) HB 315 mm. (**e**) HB 515 mm.

**Figure 5 materials-15-03757-f005:**
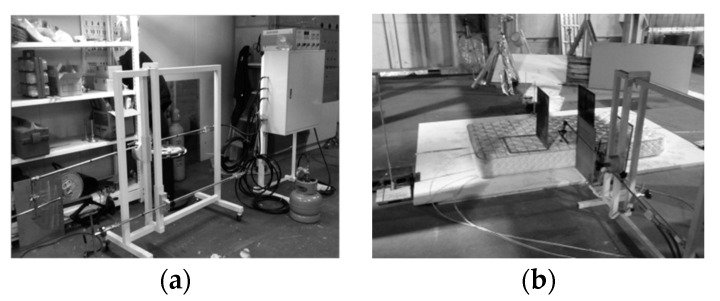
Burner’s status by ISO 12949. (**a**) Shape of burners. (**b**) Configuration of the test facility.

**Figure 6 materials-15-03757-f006:**
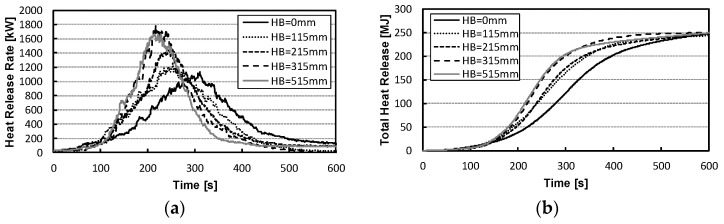
HRRs and THRs of the mattresses under different installation conditions. (**a**) HRR. (**b**) THR.

**Figure 7 materials-15-03757-f007:**
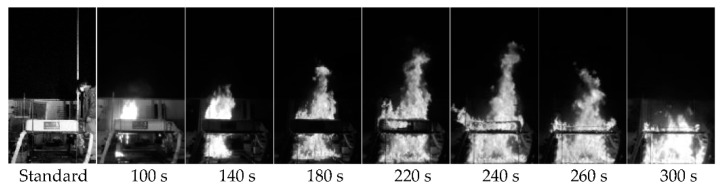
Flame-height images (HB 515 mm).

**Figure 8 materials-15-03757-f008:**
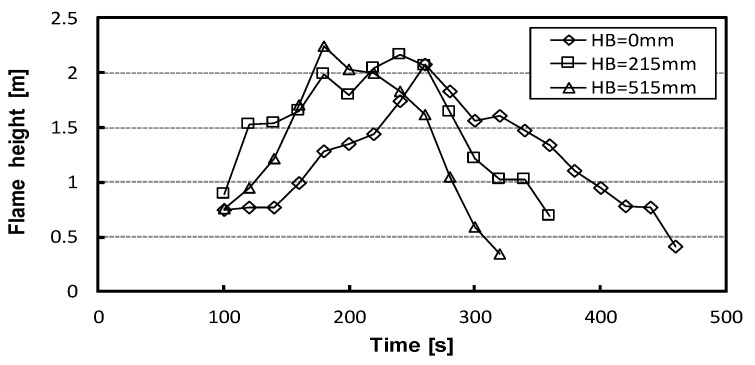
Experimental results of average flame height.

**Figure 9 materials-15-03757-f009:**
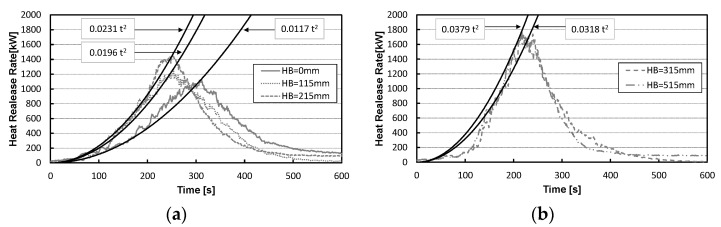
The fire growth rate of the mattress in open space. (**a**) 0–215 mm. (**b**) 315–515 mm.

**Figure 10 materials-15-03757-f010:**
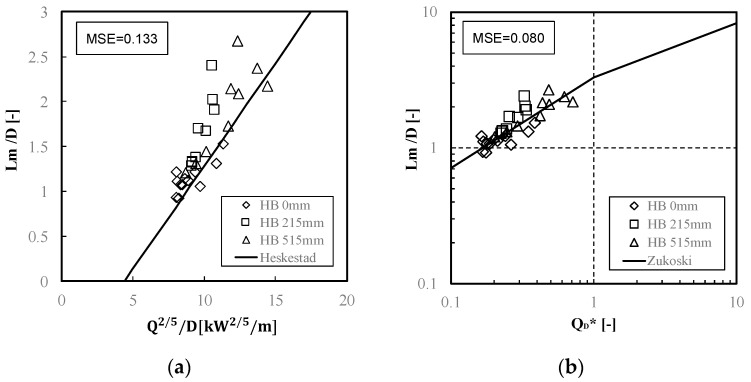
Comparisons of experimental data and two major equation models in the average flame height. (**a**) Heskestad’s formula (Equation (2)). (**b**) Zukoski’s formula (Equation (3)).

**Table 1 materials-15-03757-t001:** Values for *α* and *t_g_* for different classifications of fire growth in NFPA 72.

Classification of Fire Growth in NFPA 72	Fire Growth Time (*t_g_* (s))	Fire Growth Rate (*α* (kW/s^2^))
Slow	600	0.00293
Medium	300	0.0117
Fast	150	0.0469
Ultra-fast	75	0.1876

## Data Availability

Not applicable.

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
