# Peer review of "Fire Properties of Bed Mattresses Focusing on the Fire Growth Rate and Flame Height"

_materials, 2022, doi:10.3390/ma15113757_

Round 1

Reviewer 1 Report

The paper presents the results of a fire experiment on a mattress. The presentation of the experiment and the presentation of the results are, in my opinion, adequate.

The weakness of the paper is the concept, as it is a technical paper rather than a scientific one.

The content is more related to different standards than to the valorization of scientific knowledge.

There are two scientific papers listed in the literature, the rest are standards, conference proceedings and books.

In my opinion, the experiment report itself could not be a scientific paper, but the data obtained from the experiment could be sufficient for a dissertation and could represent a set of scientific hypotheses that could be investigated and presented in the paper.

Focus also in a wider review of the literature.

Reviewer 2 Report

In my opinion, the presented article is clearly presented. The authors prepare full-scale experiments and show results that are not only useful for a scientific audience but also for a wider audience.

Reviewer 3 Report

The paper deals with an interesting subject, and gives useful data.

However, in my opinion one point should be improved: the "Conclusions" are not really conclusions, but just a summary of the main results. Do not have the authors some specific proposals, drawn from those results,  for increasing the safety of those mattresses?

And one quite specific comment: a "Notation" section would help the reader.

Reviewer 4 Report

In the current article, an attempt is being made to evaluate the effect of bed matrices installation height on fire attributes (growth rate, the heat release rate, and the flame height), based on experimental results. I find the research interesting, but additional analysis is required to support the conclusions. My comments are provided below.

General Comments

The authors use the “significance” word to evaluate the potential differences on heat release (and consequently on fire growth) caused by installation heights (L144). In addition, according to the authors, “no significant difference in the average flame height 159 from ignition to 60 s” (L159 – 160). I suggest the authors use a simple statistical analysis (at 0.05 significance level) in order to support the results and to increase the scientific merit of the manuscript. I also suggest authors add a relevant paragraph (2.5.3) describing (briefly) the analysis that will be followed. Provided that the results of the statistical analysis are still unknown, I recommend major revision prior to publishing.

Specific comments

L12-13 Please improve the sentence.

L22-23 The keywords are already mentioned in the title.

L33-34 I suggest the authors transfer the sentence to the last paragraph of the introduction section.

L37 Please add the relevant citation (after high)

L44-47 I suggest the authors transfer the sentence to the last paragraph of the introduction section

L53 Please add Heskestad and Zukoski's reference.

L144 Please see general comments

L159 Please see general comments

Round 2

Reviewer 3 Report

In my opinion the paper can be accepted.

Reviewer 4 Report

The authors followed the recommended suggestions and the manuscript has improved significantly.